# Prevention of Overhead Shoulder Injuries in Throwing Athletes: A Systematic Review

**DOI:** 10.3390/diagnostics14212415

**Published:** 2024-10-30

**Authors:** Ayrton Moiroux--Sahraoui, Jean Mazeas, Numa Delgado, Cécile Le Moteux, Mickael Acco, Maurice Douryang, Andreas Bjerregaard, Florian Forelli

**Affiliations:** 1Orthosport Rehab Center, 95330 Domont, France; jeanmazeas@gmail.com (J.M.); mickaelacco@gmail.com (M.A.); fforelli@capio.fr (F.F.); 2Orthopaedic Surgery Department, Clinic of Domont, Ramsay Healthcare, @OrthoLab, 95460 Domont, France; 3Physiotherapy School, Institut Paramédical des Métiers de la Rééducation (IPMR), 58000 Nevers, France; cecile.lemoteux@ipmr-nevers.fr; 4Rehab Center, 21320 Pouilly-en-Auxois, France; numadelgado.mk@gmail.com; 5Department of Physiotherapy and Physical Medicine, University of Dschang, Dschang P.O. Box 96, Cameroon; douryangmaurice@gmail.com; 6Rehabilitation Department, Aspetar Orthopaedic and Sports Medicine Hospital, Doha 29222, Qatar; andreas_bjerregaard@yahoo.dk; 7SFMK Lab, 93380 Pierrefite sur Seine, France

**Keywords:** shoulder injuries, kinetic chain, baseball, overhead sports, prevention, rehabilitation

## Abstract

(1) Background: Shoulder pathologies are mostly found in overhead sports. Many risk factors have been identified, in particular a deficit in the kinetic chain. The aim of this review was to find out whether prevention by strengthening the kinetic chain can have an impact on the rate of shoulder injury in overhead pitching athletes. (2) Methods: A systematic review of the literature was carried out, including studies on the role of the kinetic chain in the prevention of overhead athletes. The studies used were works published over the last 10 years searched on PubMed, Cochrane Library, PEDro and Science Direct. They were also analyzed by methodological quality scales: the PEDro scale and the Newcastle–Ottawa scale. (3) Results: Eight studies met the inclusion criteria. The studies analyzed revealed a significant correlation between the use of the kinetic chain and the prevention of shoulder injuries, associating factors such as muscle strength, physical performance in tests (CMJ, FMS), static and dynamic balance and the ability to transfer energy from the lower to the upper body. (4) Conclusions: It is important to integrate core stability work and lower limb strengthening to minimize excessive stress on the shoulder complex, while optimizing force production and performance.

## 1. Introduction

Shoulder pathologies occur mainly in overhead sports. In baseball, for example, 12% to 19% of injuries are localized to the shoulder, while in swimming, shoulder injuries are estimated at between 23% and 38% in any given year [1,2,3]. There are several risk factors for shoulder injuries in overhead sports, such as range of motion, rotator cuff muscle weakness and training load [4,5,6]. A deficit in the kinetic chain is a risk factor for upper-limb injuries. Ball throwing is a complex movement described by most authors as occurring in six phases: ascent, stride, arm cocking, arm acceleration, arm deceleration and follow-through [7,8]. The pitching motion is part of a more global movement passing through all these phases, using complex kinetic chains to produce maximum force [9]. At maximum speed, they all turn towards the target. For professional baseball pitchers, pelvic rotation is 600°·s^−1^, starting before trunk rotation. The pelvis essentially pulls the trunk toward the target, opening the way. The trunk rotates at 1000°·s^−1^, pulling the shoulder toward the target. At maximum speed, the shoulder rotates at around 7000°·s^−1^, pulling the elbow towards the target, which then articulates towards the target at a speed of around 4000°·s^−1^ on the already fast-moving humerus [10]. This kinetic chain enables greater force to be developed and therefore the ball to be thrown more quickly. In human throwing, the winding and stride phases represent the loading of the whip. Potential energy is stored when the arm is loaded. Once late cocking has taken place, the direction of shoulder rotation changes abruptly from external to internal. As with the whip, this is a violent transition that subjects the shoulder and elbow to great stress. At this stage, the anterior capsule of the shoulder and the ulnar collateral ligament (UCL) of the elbow are under maximum stress. Thus, anterior shoulder pain is typically related to instability or anterior labrum pathology, and medial elbow pain at this stage is most likely due to LUC pathology. When the ball is released, there is less stress on the anterior cruciate ligament, and the pronator flexor group is active as the pitcher gives spin to the ball. Medial pain during this phase is more often due to pathology of the common flexor tendon. During deceleration, the posterior muscles of the rotator cuff are subjected to high eccentric loads, and posterior shoulder pain during deceleration may indicate pathology of the teres minor and/or infraspinatus [11]. In a normally functioning kinetic chain, the legs and trunk are the engine of strength development and the stable proximal base of distal mobility. This link develops 51% to 55% of the kinetic energy and force supplied to the hand, creating the angular momentum between the back leg and the front leg to push the arm forward. The shoulder produces only 13% of the total kinetic energy [12]. The aim of this systematic review was to identify ways of reducing the risk of shoulder injury through the kinetic chain in overhead athletes.

## 2. Materials and Methods

### 2.1. Search Strategy

We carried out a literature review that included studies on the role of the kinetic chain in the prevention and management of overhead athletes. We used 4 English-language databases: PubMed, Cochrane Library, PEDro, Science direct. MeSH keywords were identified to create our search equations: (“Shoulder injuries”; “shoulder”; “sport”; “sports”; “prevention”; “baseball”; “pitcher”; “baseball pitchers”; “pitching”; ‘injury prevention”; “kinetic chain”; “overhead athletes”; “overhead sports”; “rehabilitation”; “throwing”; “throwing injuries”; “functional movement screen”). In these search bases, we entered different search equations (“functional movement screen” OR “kinetic chain”) AND “overhead sports” AND “shoulder”; (“functional movement screen” OR “kinetic chain”) AND “prevention” AND “shoulder”; “Shoulder injuries” AND (“sports” OR “sport”) AND “prevention”; “Shoulder injuries” AND “baseball” AND “kinetic chain”; “Overhead athletes” AND “ injury prevention” AND “kinetic chain”) to best meet our needs. Relevant studies were identified by the lead author by systematically searching four online databases. All articles were downloaded and transferred to Zotero v6.0.13 management platform. Cross-referenced and any duplicates were deleted before the selection criteria were applied.

### 2.2. Study Selection

We searched for studies over the last ten years, between 1 January 2014 and 31 December 2023. Eligibility criteria for the inclusion of articles in our literature review were defined. Inclusion criteria included aspects such as study population (athletes practicing throwing sports), the intervention (role of kinetic chain in shoulder injuries), comparison with a non-kinetic-chain-based treatment or absence of comparison, judgment criteria (incidence of shoulder injuries or efficacy of kinetic chain treatment), study type (randomized clinical trials or other trials of inferior methodological value), language (English) and full accessibility of the article. On the other hand, certain criteria excluded articles, such systematic reviews, meta-analyses and book chapters or non-compliance with the specified language. When analyzing the articles, we extracted various data from the included studies. We used a flow chart according to the Preferred Reporting Items for Systematic reviews and Meta-Analyses (PRISMA) 2020 statement (Figure 1) [13].

We asked ourselves: How can the risk of shoulder injury via the kinetic chain be reduced in overhead athletes?

### 2.3. Data Extraction and Quality Assessment

Extracted data, conducted by AMS and ND, included study type, number of participants, participant characteristics, interventions used to treat and/or prevent shoulder injuries and study results. We used two types of methodological scale, namely the PEDro scale or the Newcastle–Ottawa scale (NOS), to determine the quality assessment of our included articles. These methodological scales have enabled us to qualify the validity of our results. PEDro scale assessed randomized control trials and the NOS evaluated the cohort and case–control studies [14,15]. The PEDro scale was used for the study of Suzuki K et al. [16] and the study of Stig H Anderson et al. [17]. The NOS was used for the other six studies. We followed the Preferred Reporting Items for Systematic Reviews and Meta-Analyses (Appendix A).

## 3. Results

After deleting duplicates, we read 30 articles. Twenty-two of them did not correspond to certain elements of our Population, Intervention, Comparison and Outcome (PICO) criteria (target population, intervention, etc.) (Table 1). We therefore selected eight for inclusion in our analysis. Eight studies were included in this systematic review (two case–control studies (25%), one cross-sectional study (12.5%), two randomized controlled trials (25%), one pilot study (12.5%), one longitudinal study (12.5%) and one cohort study (12.5%)). Studies focus on the kinetic chain and its role in the prevention plan for throwing sports.

The reviewed studies provided compelling evidence regarding the relationships between physical assessments, injury prevalence, risk factors, and the effectiveness of intervention programs in athletic populations (Table 2 and Table 3).

### 3.1. Identifying Risk Factors (2 Studies; n = 25%)

Sekiguchi et al. [19] revealed a noteworthy prevalence of shoulder and/or elbow pain among athletes, documented at 24.8%. This included 17.3% reporting shoulder pain alone, 6.6% experiencing combined elbow and shoulder pain and 8.4% with low back pain. Notably, among athletes with shoulder or elbow pain, 61.2% also reported low-back pain, and 51.9% reported knee pain, indicating a significant comorbidity that warrants attention (*p* < 0.001). The presence of low-back and knee pain was statistically linked to an increased risk of upper limb injuries, with adjusted odds ratios (ORs) demonstrating a strong association: 4.23 (*p* < 0.001) for low-back pain and 3.11 (*p* < 0.001) for knee pain. Furthermore, in Endo and Masaaki Sakamoto [20], the data suggested that pitchers experienced a higher prevalence of shoulder problems (30.4%) compared to non-pitchers (21.7%) (*p* < 0.001), emphasizing the need for targeted injury prevention strategies within this subgroup of athletes.

### 3.2. Functional Movement and Performance (3 Studies; n = 37.5%)

Andrew M Busch et al. [18] indicated that poor performance in the Functional Movement Screen (FMS) was significantly associated with an increased likelihood of developing overuse injuries during the preseason. Specifically, athletes with low FMS scores exhibited adjusted Odds Ratios (ORs) = 5.14 (*p* = 0.03) and 3.73 (*p* = 0.03) for experiencing overuse symptoms, depending on whether grade and position were controlled. Kenta Suzukii et al. [16] show that FMS scores improved significantly after a 12-week training program, highlighting the effectiveness of structured physical training. However, by 24 weeks, no significant differences in FMS scores were observed between the intervention and control groups, suggesting that continuous monitoring and training may be necessary to sustain improvements in functional movement. Conversely, poor performance in the Selective Functional Movement Assessment (SFMA) also correlated with increased injury risk during both preseason and competitive phases, reinforcing the importance of these assessments in predicting athlete health. Additionally, in Hyeyoung Kim et al. [22], specific physical training (SPT) yielded statistically significant increases in isokinetic muscle strength for internal rotation at higher angular velocities (400°·s^−1^) following training, with male athletes improving from 51.75 ± 4.50 to 66.75 ± 6.65 and female athletes from 48.00 ± 4.24 to 60.00 ± 4.24 (*p* < 0.05). Improvements were also noted in FMS scores after SPT, with male athletes’ scores increasing from 15.50 ± 1.00 to 16.25 ± 1.71 and female athletes’ scores increasing from 15.00 ± 0.00 to 17.50 ± 0.71 (*p* < 0.05). These findings highlight the potential of targeted intervention programs to enhance athletic performance while concurrently reducing injury risk.

### 3.3. Intervention Efficacy (1 Study; n = 12.5%)

The implementation of the OSTRC Shoulder Injury Prevention Program by Stig Haugsboe Andersson et al. [17], significantly reduced shoulder issues among handball players. The intervention group experienced a 28% lower risk of shoulder problems during the competitive season, demonstrating the program’s effectiveness in mitigating injury risk.

### 3.4. Balance and Flexibility (2 Studies; n = 25%)

A significant correlation was identified by Davong D Phrathep et al. [23] between contralateral hamstring flexibility and total throwing shoulder mobility, as evidenced by a positive linear correlation coefficient (r = +0.3928). This suggests that increased flexibility in the hamstrings may facilitate greater shoulder range of motion, potentially aiding in athletic performance and reducing injury risk. Furthermore, Radwan et al.’s [21] balance measurements indicated that players in the experimental group exhibited significantly lower balance scores than those in the control group, pointing to a need for targeted balance training interventions to enhance stability and performance in athletes.

### 3.5. Muscle Strength Assessments (1 Study; n = 12.5%)

For Hyeyoung Kim et al. [22], isokinetic muscle strength assessments revealed no significant changes in internal rotation strength at 240°·s^−1^; however, a significant increase was documented at 400°·s^−1^ following specific training. The absence of statistically significant differences in external rotation strength at both tested speeds suggests a potential area for further investigation and targeted training to enhance this aspect of muscle performance. These findings underscore the importance of tailored training interventions to improve specific muscle strength profiles among athletes.

Overall, the findings underscore the critical relationships between functional assessments, targeted interventions, and injury prevalence in athletic populations. The significant associations between lower limb flexibility, shoulder mobility, and injury risk highlight the need for integrated training approaches that encompass flexibility, strength, and balance. Future research should continue to explore these dynamics, focusing on long-term outcomes of intervention strategies to better inform injury prevention and performance enhancement practices in sports settings.

### 3.6. Quality Data

We have two randomized control trials in this review, that of Stig Haugsboe Andersson et al. [17] and that of Kenta Suzuki et al. [16]. The former has a score of 5/10 on the PEDro scale. The second gives us a score of 7/10. We used the NOS scale for case–control and cohort studies. Radwan A et al.’s [21] study has a score of 9/9. Finally, the case–control study by Hyeyoung K et al. [22] scores 9/9. Next, we used the same scale for cohort studies. We start with the study by Sekiguchi T et al. [19], which scores 7/9. Andrew M Busch et al.’s [18] study scores 8/9. Finally, Endo Y et al. [20] scored 8/9. The main biases found were failure to blind therapists to the treatment followed by the subjects. Some of them used self-evaluations or declarations to evaluate certain criteria. Follow-up was not always long enough to have a real effect on the endpoint.

For pilot studies, there is no validated methodological quality scale. However, we can analyze the study with the biases found in the methodological scales of other types of studies. The study by Davong D Phrathep et al. [23] involved subjects from the same community. The study controls for the most important factor. All subjects were followed up with. On the other hand, the study does not take place over an extended time frame, so we have no follow-up over an extended period. In addition, the study was not blinded.

## 4. Discussion

The aim of this systematic review was to identify ways of reducing the risk of shoulder injury through the kinetic chain in overhead athletes. The studies analyzed reveal a significant correlation between the use of the kinetic chain and the prevention of shoulder injuries, linking factors such as muscular strength, physical performance, balance and the ability to transfer energy from the lower to the upper body. More specifically, the results indicate that musculoskeletal injuries and pain are major problems in sport, with a particular focus on shoulder, elbow, low back, and knee injuries. Sport-specific adaptations can lead to chronic shoulder pain, influencing biomechanics and movement strategies during activities [24]. Poor performance on FMS tests could be used to predict sports injuries [25]. In addition, studies have identified correlations between physical performance, muscle strength and FMS scores. Targeted interventions such as specific training programs can effectively reduce the risk of injury, particularly in athletes practicing sports requiring overhead movements, such as strength training or proprioception training [26]. Studies also highlight the importance of flexibility, mobility, strength and balance in preventing injury and improving athletic performance [19,20,21,23]. Aside from interventions based on direct shoulder strengthening and treatment, a variety of exercises, including isolated lower limb movements and movements involving the kinetic chain, with sets of three and a number of repetitions around 10 to activate muscles, are recommended for optimal effectiveness in prevention for baseball pitchers [27]. Exercise programs focusing on core stability, lower-limb strengthening and throwing-movement-specific exercises are essential to prevent injury and improve performance [28]. It is also important to include exercises aimed at improving shoulder mobility to reduce muscle tension [29]. Future research into injury prevention in overhead throwers needs to focus on several areas to better understand injury mechanisms and develop more effective prevention strategies. The development of personalized interventions, considering the individual characteristics of each athlete, could strengthen weak points, correct muscular imbalances and improve throwing technique.

### Limitations

We had to pay particular attention to managing potential biases that could compromise the validity and credibility of our work. In addition, publication bias may also influence our review. Studies with positive results tend to be published more readily than those with negative results, creating an imbalance in the representation of available data. This can distort our understanding of the current state of knowledge in this field. In addition, it is important to be vigilant about citation bias. The main issue here, above all, is that having only a single reviewer introduces the risk of bias, whether intentional or unintentional. This becomes even more problematic given that the reviewer is aware of the study’s objectives, which could consciously or unconsciously influence the selection and interpretation of the data. This involves an exhaustive search of the literature, examining not only the most accessible sources, but also actively searching for studies in lesser-known journals or specialized databases. In addition, a critical evaluation of the included studies is essential. This involves analyzing research methods, samples used, outcome measures and conclusions drawn. Manuscripts have been reviewed by a single author due to time or resource constraints. Likely biases within the reviewed literature could include selection bias, publication bias and confirmation bias. Additionally, biases could stem from differences in study design, sample sizes or inconsistent methodologies across the studies reviewed. Finally, full transparency in the selection and analysis methods used is crucial to establishing the credibility of our review. By working in collaboration with many expert and methodological directors, we can ensure that our literature review is rigorous, objective and informative.

## 5. Conclusions

This work highlights the importance of strengthening the kinetic chain in the prevention of shoulder injuries in overhead athletes. Here are a few key points:Shoulder or elbow pain often coincides with lower back and knee pain, increasing the risk of upper limb injuries. Pitchers have a higher rate of shoulder problems.Poor FMS results increase the risk of overuse injuries. Training improves FMS and muscular strength, but it must be continuous.The OSTRC program reduces shoulder injuries by 28% in handball players.Improved hamstring flexibility improves shoulder mobility, and balance training is essential for injury prevention.Training improves internal rotation strength, but more attention needs to be paid to external rotation.

It is important to integrate trunk stability work and lower limb strengthening to minimize overstressing of the shoulder complex, while optimizing force production and performance. However, there is still a great deal of research to be conducted before we have a better understanding enabling the implementation of the protocol.

## Figures and Tables

**Figure 1 diagnostics-14-02415-f001:**
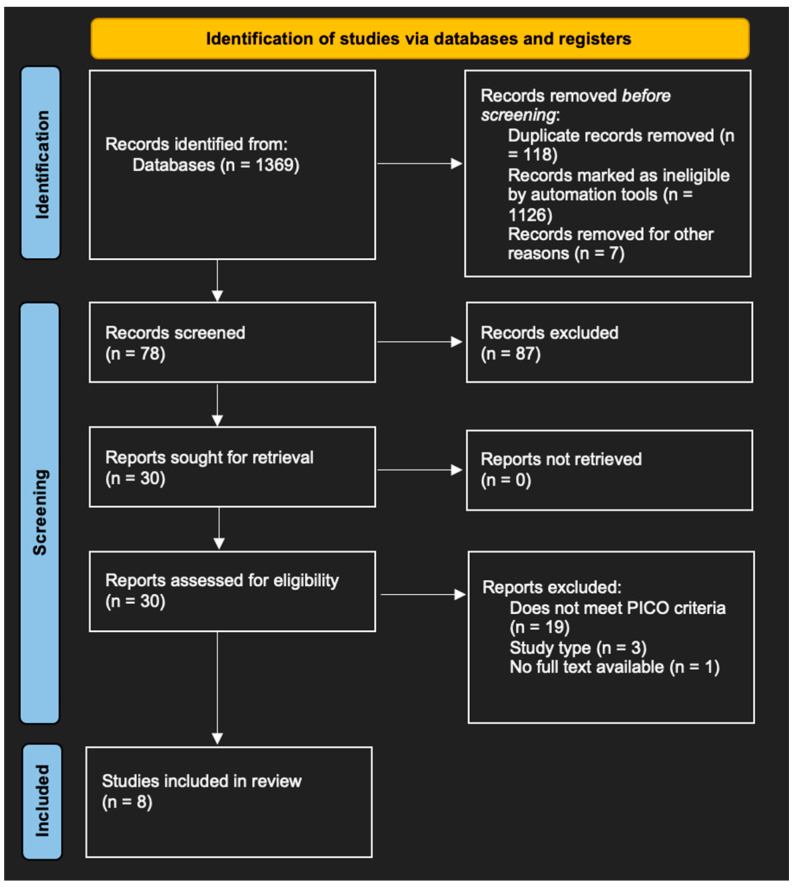
Flow chart.

**Table 1 diagnostics-14-02415-t001:** Table of PICO criteria for articles used in the literature review.

Articles	Author	Population	Intervention	Comparison	Outcomes
**Relationship of preseason movement screens with overuse symptoms in collegiate baseball players**	Andrew M Busch et al. [18]	Baseball players at university	Pre-season FMS and SFMA scores	No comparison	The number of shoulder injuries during the season.
**Youth baseball players with elbow and shoulder pain have both low back and knee pain: a cross-sectional study**	Sekiguchi T et al. [19]	Young baseball players	Questionnaire on the association of shoulder and/or elbow injuries with trunk and/or lower limb pain	Athletes with shoulder injuries uncorrelated with trunk and/or lower limb injuries	The number of participants with shoulder and/or elbow pain correlated with trunk and/or lower limb pain.
**Efficacy of Injury Prevention Using Functional Movement Screen Training in High-School Baseball Players: Secondary Outcomes of a Randomized Controlled Trial**	Suzuki K et al. [16]	High school baseball players	FMS training 4 times a week for 12 weeks	Normal training	Improvement in FMS score.
**Correlation of shoulder and elbow injuries with muscle tightness, core stability, and balance by longitudinal measurements in junior high school baseball players**	Endo Y et al. [20]	High school baseball players	Muscle tension test, SEBT and trunk endurance test	No comparison	The number of players with shoulder or elbow injuries during the throw.
**Preventing overuse shoulder injuries among throwing athletes: a cluster-randomised controlled trial in 660 elite handball players**	Stig H Anderson et al. [17]	Professional handball players	OSTRC Shoulder Injury Prevention program during warm-up 3 times a week	Normal warm-up	The prevalence of shoulder problems and substantial shoulder problems in the dominant arm and the severity score of shoulder problems reported during the season.
**Is there a relation between shoulder dysfunction and core instability?**	Radwan A et al. [21]	Overhead athletes	Functional questionnaires (KJOC and QuickDASH sports model) as well as single-leg balance test (SLBT), double straight leg lowering test (DLL), Sorensen test and modified lateral sheathing test	No comparison	To analyze the difference between healthy athletes and those with shoulder dysfunction in measures of trunk stability and to explore the relationship between measures of trunk stability and measures of shoulder dysfunction.
**Effects of 8 weeks’ specific physical training on the rotator cuff muscle strength and technique of javelin throwers**	Hyeyoung K et al. [22]	Javelin throwers	The 8-week specific physical training program (strength training; Javelin-specific training; FMS training)	Normal training	Studied changes in rotator cuff muscle strength, throwing distance and technique in javelin throwers after they had undergone specific physical training.
**Throwing Shoulder Range of Motion and Hamstring Flexibility in Adolescent Baseball Players: A Pilot Study**	Davong D Phrathep et al. [23]	Baseball players	Measure hamstring flexibility and throwing shoulder amplitude	Players without hamstring stiffness	Study the relationship between hamstring flexibility and throwing shoulder amplitude.

**Table 2 diagnostics-14-02415-t002:** Demographic information.

Articles	Author	Year	*N*	Sex	Age	Population	Sport
**Relationship of preseason movement screens with overuse symptoms in collegiate baseball players**	Andrew M Busch et al. [18]	2017	135	Male	20.1 ± 2.0 years	Athletes	Baseball
**Youth baseball players with elbow and shoulder pain have both low back and knee pain: a cross-sectional study**	Sekiguchi T et al. [19]	2018	1582	Male and female	11.0 ± 1.0 years	Athletes	Baseball
**Efficacy of Injury Prevention Using Functional Movement Screen Training in High-School Baseball Players: Secondary Outcomes of a Randomized Controlled Trial**	Suzuki K et al. [16]	2022	71	Male	16.0 ± 1.0 years	Athletes	Baseball
**Throwing Shoulder Range of Motion and Hamstring Flexibility in Adolescent Baseball Players: A Pilot Study**	Davong D Phrathep et al. [23]	2023	15	Male	16.0 ± 2.0 years	Athletes	Baseball
**Preventing overuse shoulder injuries among throwing athletes: a cluster-randomised controlled trial in 660 elite handball players**	Stig H Anderson et al. [17]	2017	660	Male and female	22.5 ± 4.0 years	Athletes	Handball
**Is there a relation between shoulder dysfunction and core instability?**	Radwan A et al. [21]	2014	61	Male and female	19.3 ± 1.1 years	Athletes	American football, swimming, water polo, lacrosse, baseball, softball, field throwing, basketball
**Correlation of shoulder and elbow injuries with muscle tightness, core stability, and balance by longitudinal measurements in junior high school baseball players**	Endo Y et al. [20]	2014	39	Male	13.5 ± 0.5 years	Athletes	Baseball
**Effects of 8 weeks’ specific physical training on the rotator cuff muscle strength and technique of javelin throwers**	Hyeyoung K et al. [22]	2014	10	Male and female	22.0 ± 1.15 years	Athletes	Javelin throwing

**Table 3 diagnostics-14-02415-t003:** Outcomes measured.

Articles	Author	Assessments Used	Data	*p*-Value
**Relationship of preseason movement screens with overuse symptoms in collegiate baseball players**	Andrew M Busch et al. [18]	-FMS score -SFMA Score	-FMS performance is associated with an increased likelihood of experiencing overuse symptoms during the preseason (OR of 5.14). -SFMA performance is linked to a higher risk of overuse symptoms during the preseason (OR of 6.10). -During the competitive season, poor SFMA performance is associated with a significantly higher risk of overuse symptoms (OR = 17.07).	- 0.03 - 0.03 - 0.03
**Youth baseball players with elbow and shoulder pain have both low back and knee pain: a cross-sectional study**	Sekiguchi T et al. [19]	-Pain	Knee pain was also significantly associated with shoulder pain (OR = 3.31) and combined elbow and shoulder pain (OR = 3.52).	<0.01
**Efficacy of Injury Prevention Using Functional Movement Screen Training in High-School Baseball Players: Secondary Outcomes of a Randomized Controlled Trial**	Suzuki K et al. [16]	-Number of injuries -Time lost due to injury -FMS score	-Non-contact injuries alone also significantly decreased in the intervention group (3 injuries) compared to the control group (10 injuries) (power = 0.99). -After 12 weeks, the intervention group had a significantly higher total FMS score (17.5 ± 1.5) than the control group (14.7 ± 2.3).	- 0.02 - <0.01
**Throwing Shoulder Range of Motion and Hamstring Flexibility in Adolescent Baseball Players: A Pilot Study**	Davong D Phrathep et al. [23]	-Shoulder ROM -Hamstring ROM	A positive linear relationship was found between contralateral hamstring flexibility and total throwing shoulder mobility, with a correlation coefficient (r) of +0.3928.	
**Preventing overuse shoulder injuries among throwing athletes: a cluster-randomised controlled trial in 660 elite handball players**	Stig H Anderson et al. [17]	-OSTRC Shoulder Injury Prevention program	Players in the intervention group had a 28% lower risk of reporting shoulder problems during the competitive season compared to those not participating in the program.	<0.05
**Is there a relation between shoulder dysfunction and core instability?**	Radwan A et al. [21]	-Sorensen test -DLL test -Right and left side plank tests -Right and left SEBT	The experimental group had significantly lower balance than the control group, with mean scores of 10.14 ± 5.76 and 18.98 ± 15.22, respectively.	0.038
**Correlation of shoulder and elbow injuries with muscle tightness, core stability, and balance by longitudinal measurements in junior high school baseball players**	Endo Y et al. [20]	-Muscle tension test -SEBT -Trunk endurance test	Compared to the non-painful group, players in the painful group experienced a significant increase in tension in the shoulder internal rotators, quadriceps of the axial leg, and hamstrings of the axial leg.	<0.01
**Effects of 8 weeks’ specific physical training on the rotator cuff muscle strength and technique of javelin throwers**	Hyeyoung K et al. [22]	-FMS score -Isokinetic -Kinematic analysis	-A statistically significant increase in muscle strength was noted at 400°·s^−1^ after SPT: -Male athletes: before 51.75 ± 4.50 vs. after 66.75 ± 6.65; -Female athletes: before 48.00 ± 4.24 vs. after 60.00 ± 4.24 (*p* < 0.05). -A statistically significant increase in FMS scores was observed after SPT: -Male athletes: before 15.50 ± 1.00 vs. after 16.25 ± 1.71; -Female athletes: before 15.00 ± 0.00 vs. after 17.50 ± 0.71.	- <0.05 - <0.05

Note: FMS = Functional Movement Screen; SFMA = Selective Functional Movement Assessment; OR = Odds Ration; ROM = Range Of Motion; OSTRC = Oslo Sports Trauma Research Center; DLL = Double Straight Leg Lowering Test; SEBT = Star Excursion Balance Test; KJOC = Kerlan–Jobe Orthopaedic Clinic; SPT = Specific Physical Training.

## Data Availability

The original contributions presented in the study are included in the article/Appendix A, further inquiries can be directed to the corresponding author.

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
