# Peer review of "Prevention of Overhead Shoulder Injuries in Throwing Athletes: A Systematic Review"

_diagnostics, 2024, doi:10.3390/diagnostics14212415_

Round 1
Reviewer 1 Report
Comments and Suggestions for Authors
Introduction First paragraph: This section contains many statements without clear attribution to a source. For example, lines 39 to 45 all seem to come from reference 10, but this is not entirely clear. I would prefer to see the source attribution early and clear language linking further statements to that source. Also, statements are presented without specification of the type of athlete the data describes (professional, college, high school, recreational….).
Methods: Please note the quality assessment metric used for each study in your data tables. Also, I do not see any quality data presented in the Results.
Results: The text of the results is difficult to follow, as it is not organized around a set of central questions or topics. In many areas it seems that the text states the same information as the data tables. I would strongly encourage the authors to revise the Results to follow a format that organizes the information addressing central questions or topics of utility to the reader, rather than listing information from the sources without structure.
Discussion: The statement “Studies analyzed reveal a significant correlation between the use of kinetic chain and the prevention of shoulder injuries” is not well supported with the Results. Reorganization of data in the results section would help support the authors case.
The statement “Targeted interventions…” is also not well supported. Ideally a review paper should be structured such that key questions are identified in the Methods, answers are sequentially presented in the Results, and then arguments in the Discussion will be strongly supported. This did not occur in the current manuscript.
Line 18: In discussing exercises, statements should also be made describing the frequency and length of training, along with the athletic type of population.
Limitations: Why were manuscripts only reviewed by a single author? Why wasn’t data on bias provided in the paper? What were biases that likely existed within the reviewed literature?
Author Response
Thanks to the reviewer for taking the time to give us feedback. I enclose the answers in a Word document.

Reviewer 2 Report
Comments and Suggestions for Authors
Thank you for the opportunity to review this manuscript. Although this paper discusses an important subject matter, your manuscript would benefit from revisions and a thorough editing as there are numerous grammatical mistakes.
- - Please include a research question or define your PICO in your methods
- - In your Figure 1, please replace Reason 1,2,3 with actual reasons
- - It seems you used both the PEDro and the NOS scale. You also didn’t utilize any other scales. Therefore, the first sentence of your methodological quality needs to be adjusted to reflect this.
- - Simple presenting your quality assessment in your Table is not sufficient. Your manuscript requires a more thorough verbal synthesis of the results from your quality assessment. It would help if you attached references to your findings to identify which studies failed to blind or used self-report. Also, why is the first sentence written in future tense? Similar to the comment above, state what you actually did. i.e. “The PEDro and NOS scales were used to ….” Unless you used another scale that was not reported on in your manuscript
- - Be consistent in your use of numbers. At times you write out numbers <10, at times you use the numeric value. Also, spell out numbers that begin sentences (i.e. page 3, “22 of them did not correspond”
- - Please clarify who and how many carried out the screening?
- - I would suggest splitting your Table 1 into 2 tables. In its current form, there is too much text. Your Table 1 would appear clearer if you split up column 1 into separate columns, and also added sample size and population descriptors. You can keep Column 2 as well (Study Objectives). However, I would create a separate column for Outcome Measures and Assessment and then include actual scores/results as you have verbally synthesized in your results of individual studies section. The goal of your results is not only to synthesize but to find trends. This data should be grouped together by outcome measure or theme such as FMS. When you report all results individually and verbally, it becomes hard to surmise any conclusions. This would also help to support your summary of the findings.
- - Are you able to specific or recommend any strengthening exercises for the kinetic chain’ or stability exercises for the lower limb etc. You state the following conclusions: “ This work highlights the importance of strengthening the kinetic chain in the prevention of shoulder injuries in overhead athletes. It is important to integrate trunk stability work and lower limb strengthening to minimize overstressing of the shoulder complex, while optmizing force production and performance.” However, this information is not new. The results of your systematic review should help to improve knowledge outside of what is already known.
- - Number 12 & 15 on your PRISMA checklist, should NOT be N/A. Risk of bias is an important part of a systematic review. You completed the PEDro Scale which can help you to discern bias in your included articles. This needs to be reported.
Comments on the Quality of English LanguageYour manuscript would benefit from a thorough editing as there are numerous grammatical mistakes.
Author Response

(The authors gave the same response as above.)

Round 2
Reviewer 2 Report
Comments and Suggestions for Authors
You have addressed some of the concerns with respect to your revisions. However, as previously stated not only by me but from the other reviewer as well, your results section contains a lot of information and the significance of your findings can not be easily ascertained in its current format. Simply adding "pain and kinetic chain" and "prevention and kinetic chain" does not provide clarity.
It feels as though your written results section and Table 1 are reversed. Your Table 1 is too full of too much text and contains too much information; whereas your written results contains too much data. Your table is supposed to provide added benefit and clarity to your written results section by organizing and summarizing the data from your included articles. Also, you have text in your table that provides opinions and conclusions - which should actually be described in your results written/discussion.
Your results would be clearer if you presented all data points such as outcome measures with corresponding results (e.g. OR or ROM) so that the reader can easily summarize what was presented in your included articles. If you do not want to split your table, you need to at least add the abovementioned column and remove some of the text from the table and add them to the actual text in the results section.
Comments on the Quality of English Language
I would also suggest a thorough external edit of your manuscript to improve the quality of language in this paper. Some of your language lacks the formality that a research paper should possess.
Author Response

(The authors gave the same response as above.)

Round 3
Reviewer 2 Report
Comments and Suggestions for Authors
Thank you for your revisions. Overall, the written portion of the paper reads better and is more organized. However, you still have not made adjustments to Table 2. You have multiple outcomes and information combined into each column. Additionally, not all of your columns present the same category of information. I would suggest splitting out your demographic information (author, year, n, sex, gender, age, population, sport) and then creating another table that has outcome measure (i.e. FMS, pain) and a separate column for the data (i.e. OR) and another column for the p-values. Additionally, what PROM was used to report or objectify pain? When possible, these PROMs should also be included in your outcome measure column. This will help you and the reader to easily synthesize the results in order to assess the impact of your results.
Author Response
Thank you for the advices. We've split table 2 into two new tables according to your recommendations. We hope you find it convenient
Round 4
Reviewer 2 Report
Comments and Suggestions for Authors
Your tables look much better. Good luck with your publication.